# Genome-Wide Identification and Expression Analysis of the Stearoyl-Acyl Carrier Protein Δ9 Desaturase Gene Family under Abiotic Stress in Barley

**DOI:** 10.3390/ijms25010113

**Published:** 2023-12-21

**Authors:** Mingyu Ding, Danni Zhou, Yichen Ye, Shuting Wen, Xian Zhang, Quanxiang Tian, Xiaoqin Zhang, Wangshu Mou, Cong Dang, Yunxia Fang, Dawei Xue

**Affiliations:** 1College of Life and Environmental Sciences, Hangzhou Normal University, Hangzhou 311121, China; ding514019@163.com (M.D.); zzz20211218@163.com (D.Z.); yeyichende@outlook.com (Y.Y.); stwen2468@163.com (S.W.); zhangxian@hznu.edu.cn (X.Z.); quanxiang@hznu.edu.cn (Q.T.); zxq@hznu.edu.cn (X.Z.); mwsdiana@163.com (W.M.); dangcong@hznu.edu.cn (C.D.); yxfang12@163.com (Y.F.); 2Zhejiang Provincial Key Laboratory for Genetic Improvement and Quality Control of Medicinal Plants, Hangzhou Normal University, Hangzhou 311121, China

**Keywords:** barley, SAD, evolution, unsaturated fatty acids (UFAs), abiotic stress

## Abstract

Stearoyl-acyl carrier protein (ACP) Δ9 desaturase (SAD) is a critical fatty acid dehydrogenase in plants, playing a prominent role in regulating the synthesis of unsaturated fatty acids (UFAs) and having a significant impact on plant growth and development. In this study, we conducted a comprehensive genomic analysis of the SAD family in barley (*Hordeum vulgare* L.), identifying 14 HvSADs with the FA_desaturase_2 domain, which were divided into four subgroups based on sequence composition and phylogenetic analysis, with members of the same subgroup possessing similar genes and motif structures. Gene replication analysis suggested that tandem and segmental duplication may be the major reasons for the expansion of the *SAD* family in barley. The promoters of *HvSADs* contained various *cis*-regulatory elements (CREs) related to light, abscisic acid (ABA), and methyl jasmonate (MeJA). In addition, expression analysis indicated that *HvSADs* exhibit multiple tissue expression patterns in barley as well as different response characteristics under three abiotic stresses: salt, drought, and cold. Briefly, this evolutionary and expression analysis of *HvSADs* provides insight into the biological functions of barley, supporting a comprehensive analysis of the regulatory mechanisms of oil biosynthesis and metabolism in plants under abiotic stress.

## 1. Introduction

The unsaturated fatty acids (UFAs) are produced through a series of desaturation and elongation processes of saturated fatty acids, which are a highly sophisticated mechanism in plants. Stearoyl-acyl carrier protein (ACP) Δ9 desaturase (SAD) is a decisive enzyme located in the plastid, catalyzing the dehydrogenation of stearic acid (C18:0) at a specific position in the fatty acid chain to form a double bond for the first step of desaturation, resulting in the formation of monounsaturated fatty acid (C18:1) [1,2]. Hence, SAD directly determines the ratio of saturated fatty acids to UFAs in plant oils [3]. Oleic acid, as the main form of monosaturated fatty acid exported by plastids, can be further desaturated into polyunsaturated fatty acid derivatives and used as a main component of the cell membrane system in the form of phospholipids [4].

Fatty acids and their derivatives are important energy-storage substances and are the main components of cell membrane lipids in plants. They are particularly actively synthesized during seed development and have been validated in most species. *GmSAD5* is highly expressed in soybeans during the middle and late stages of seed development, consistent with the oil enrichment period. It was found to have strong selectivity for stearic acid substrates and could efficiently catalyze the biosynthesis of monounsaturated oleic acid [5]. Expression analysis showed that *GhA-SAD6* and *GhD-SAD8* in cotton are preferentially generated during endosperm development and are responsible for producing palmitoleic acid in cotton seed oil [6]. Four *SAD*-encoding genes (*FAB2*, *AAD5*, *AAD1*, and *AAD6*) in *Arabidopsis* are also transcriptionally induced in seeds, where *FAB2*, *AAD5*, and *AAD1* are involved in the formation of the embryonic corneum [7]. Interestingly, *TcSAD1* in cacao is universally expressed in all the tissues and is highly correlated with dramatic changes in fatty acid composition during seed maturation [8]. Overall, these results support that various *SADs* in these plants are intimately linked to the lipid synthesis pathway and seed development.

Fatty acids and their derivatives are extensively involved in regulating various defense pathways such as basal immunity, effector-induced resistance, and systemic resistance. Research has found that endoplasmic reticulum membranes of oilseed rape grown at low temperatures (4 °C) are rich in polyunsaturated fatty acids and have a significantly altered lipid composition. Additionally, an allele of *SAD*, homologous to *AtSAD6* (*At1g43800*), was also found to be responsive to, and up-regulated by, low temperatures in oilseed rape [9]. The overexpression of the *ScoSAD* prominently improved the freezing resistance of transgenic potato plants [10]. The transcription levels of soybean *SAD* and *FAD2-1* were markedly increased under low-temperature treatment. Additionally, the transcription level of *FAD2-1B* was also higher than that of *FAD2-1A* after 35 days of flowering [11]. Treating peanut seedlings with 250 mM of NaCl weakened the activity of ω-3 fatty acid desaturase, leading to decreased contents of UFAs, which affected the stability and fluidity of the membrane and even caused irreversible damage to the plant. On the contrary, increasing the content of UFAs contributed to the improvement of salt tolerance in *Suaeda salsa* [12,13,14]. Apart from responding positively to temperature, drought, and salt stresses, FAs also serve defense purposes by affecting hormone levels. Low oleic acid induced the accumulation of salicylic acid (SA) and activated defense responses in *GhSSI2s*-silenced cotton to resist disease, whereas the disruption of *GhSSI2s* directly activated resistance gene-dependent defenses [15]. In addition, UFAs reacted immediately with reactive oxygen species (ROS) to maintain the stability of the cytomembrane and organelle membranes by regulating ROS levels [16,17,18]. Given the important role of FAs in regulating metabolic homeostasis and ensuring normal plant growth, this study explores the bioinformatics aspects of SAD.

Barley is the fourth largest cereal crop in the world after wheat, rice, and maize [19]. The analysis and publication of the barley genome have facilitated high-resolution, precision genomics research, and studying specific gene families in barley can help elucidate the molecular genetic mechanisms of wheat crops. A significant number of gene families have been identified in barley [20,21,22,23,24,25,26], but systematic research on the *SADs* encoding FA dehydrogenase in barley has not yet been reported. This study screened the barley *SAD* family based on database data to confirm 14 members distributed on four chromosomes, with relatively conservative protein motifs. This study focuses on 14 *HvSADs* and investigates their expression under abiotic stress using bioinformatics analysis, with a view of exploring the excellent resistance function of *SADs* and providing a theoretical basis for genetic improvement and molecular breeding in barley.

## 2. Results

### 2.1. Identification and Characterization of SAD Proteins in Barley

Through homologous comparison and domain validation, 14 SADs were identified in the whole barley genome and designated as HvSAD1 to HvSAD14. The physicochemical properties of the SADs, such as the molecular weight (MW), number of amino acids (aa), theoretical isoelectric point (pI), aliphatic index, and other information, are summarized in Table 1. The average protein length of the HvSADs was 396 aa, ranging from 339 (HvSAD12) to 428 (HvSAD6) aa, while the corresponding MWs were 38–47 kDa, with an average of approximately 44 kDa. The average theoretical pI was 7.23, distributed between 5.97 (HvSAD12) and 9.03 (HvSAD7). Half of the HvSADs were acidic, and the rest were alkaline. The aliphatic index of the HvSADs ranged from 72.91 to 87.33, with an average of 79.71. Moreover, all of the HvSADs possessed negative GRAVY values (ranging from −0.177 to −0.490), indicating the hydrophilic character of the HvSADs. The subcellular localization of the proteins showed that most HvSADs were located in chloroplasts, with a few located in the cytoplasm and mitochondria, which is speculated to affect photosynthesis. The variability in the physicochemical properties of the HvSADs may reflect a diverse protein structure and function. All characteristics of the HvSADs are shown in Table 1.

### 2.2. Sequence Alignment and Phylogenetic Analysis

The sequence alignment results of the HvSADs showed that the overall amino acid similarity among proteins was 57.62%. Different colors were used to visually represent the proportion of homology, with yellow representing 100%, blue representing ≥ 75%, and purple representing ≥ 50%. From Figure 1, it can be observed that the homologous proportion of HvSADs ranging from 50% to 100% accounted for a higher proportion of the total sequence. The black line represents the region of the FA_desaturase_2 domain. The conserved motifs of Motif1, Motif2, Motif3, Motif5, and Motif6 are common components of the domain and may be an important criterion for determining whether the candidate’s SADs possess this domain (Figure 1). HvSAD12 was found to have significantly different homologous sequences compared with other members, thereby increasing the difference in the comparison results.

### 2.3. Phylogenetic Analysis of SAD Proteins

Elucidating phylogenetic relationships is essential for understanding the structure and evolution of gene families. By comparing the SADs of barley, rice, wheat, maize, *A. thaliana*, *G. hirsutum*, *S. lycopersicum*, and *G. max*, we classified HvSADs and their homologs into four groups, namely Class I–Class IV, based on the distance of the genetic relationships (Figure 2). No HvSAD member was distributed in Class I, and most of them were distributed in Class II and Class IV rather than in Class III. HvSAD1 and HvSAD14 are more closely related, and HvSAD2, HvSAD3, and HvSAD13 also belong to the same branch. As a whole, HvSADs are closely associated with homologs in wheat, rice, and maize, and most are not on the same terminal branch as cotton, tomato, and *Arabidopsis*, suggesting higher evolutionary homology with monocotyledonous plants than dicotyledonous plants. Notably, most HvSADs are closely related to AtSAD6 (At1g43800) and do not belong to the same branch as other members of *Arabidopsis*, indicating the possibility that AtSAD6 plays an important role in the evolution of SADs in connecting monocots and dicots.

### 2.4. Gene Structure and Protein Motif Analysis of the HvSADs Family

To further explore the evolutionary, functional diversity, and domain characteristics of the HvSADs, the structural distribution of exons, introns, non-coding regions, and conserved motifs was investigated (Figure 3a, Appendix A). Ten motifs were identified in HvSADs, and the results showed that the rest of the motifs, except for Motif8 and Motif9, were associated with the FA_desaturase_2 domain. HvSADs have five common motifs in this domain, whereas the distribution of other motifs is not significantly different, such as Motif7, Motif4, Motif10, and Motif8. It is noteworthy that except for HvSAD12, the other members also possess Motif4, Motif7, and Motif10 related to the domain (Figure 1 and Figure 3). In Class IV, except for a lack of Motif4 for HvSAD2, all other members have 10 predicted motifs and a concentration of exon regions (Figure 3b).

The genetic structure composition indicated that *HvSADs* have fewer introns, but the intron composition of *HvSAD4*, *HvSAD7*, and *HvSAD9* is higher compared to other members. In addition, the phylogenetic tree shows a high similarity of gene structures at the end of the same branch and a low similarity of gene structures between branches of different divisions, which may have been in adaptation to the external environment, leading to the diversity of gene families.

### 2.5. Cis-Regulatory Elements Analysis in the Promoters of HvSADs

Plants need to rapidly adapt to the constantly changing environment and ensure timely defense to support their normal growth and development. Understanding the CRE composition of the *HvSAD* promoters can help elucidate the potential factors affecting *HvSAD* expression and the regulatory pathways for participation. The results showed that common CREs (such as TATA-box and CAAT-box) are present in the promoter regions of *HvSADs*. Furthermore, there are many other functional elements (Figure 4), which can be categorized into four groups: (1) growth and development regulation elements, such as zein metabolism regulation-related element (O2-Site) and meristem expression-related element (CAT-box); (2) stress-responsive elements, such as drought-responsive element (MBS), low temperature-responsive element (LTR), and anaerobic-responsive element (ARE); (3) hormone-responsive elements, such as ABA-responsive element (ABRE), auxin-responsive element (TGA element), salicylic acid-responsive element (TCA-element), and MeJA-responsive element (CGTCA motif/TGACG motif); (4) light-responsive elements, such as Box 4, G-box, GT1-motif, and SP1. There were 33, 66, 184, and 182 elements responsible for plant growth and development, stress response, hormone response, and light response, respectively, accounting for 7%, 14%, 40%, and 39% of all CREs in the *HvSAD* promoter regions. *HvSADs* had the most hormone-responsive elements distributed in the promoter regions, and except for *HvSAD4*, they all contained ABRE elements.

The element’s response to plant hormones, light, and environmental stresses was detected in all the *HvSADs*, while only growth and development regulation elements were lacking in the *HvSAD9.* In plant growth and development, the CRE related to light response and meristem expression (CAT-box, 15) was relatively high. In hormone response, the CREs associated with ABA (ABRE, 69) and MeJA (TGACG motif, 42; CGTCA motif, 42) were the most. In the stress response, the CREs related to anaerobic response (ARE, 24) and GC motif (14) were relatively high. The above results indicated that *HvSADs* are not only induced by adversity, hormones, and light, but may participate in plant growth and development processes. The different numbers of CREs reflect the different modes in which *SADs* are regulated.

### 2.6. Chromosome Location and Gene Duplication Analysis of HvSADs

*HvSADs* are unevenly distributed on four of the seven barley chromosomes (Figure 5). Among them, there are six, four, and three genes on chromosomes 2, 3, and 5, respectively, while only one *HvSAD* is present on chromosome 7, and most *HvSADs* are located near the ends of the chromosome arm.

Tandem and segmental duplication are two forms of gene duplication events that are major forces driving the expansion of gene families and the evolution of the entire genome [27]. Collinearity analysis revealed the presence of *SAD* gene duplication in the barley genome. Based on the evolutionary relationships and the distance between homologous gene pairs, two pairs of genes were each determined to have undergone tandem and segmented duplication (Figure 6). For instance, *HvSAD2* and *HvSAD13* underwent tandem replication and are tightly arranged on the same chromosome to form gene clusters with similar sequences and functions. In the same family, *HvSAD4* and *HvSAD9* are, respectively, located at a distance of 2H and 3H, which is the result of segmental duplication. To further explore selective pressure in barley, the synonymous (Ks) and nonsynonymous (Ka) nucleotide substitution rates of four homolog pairs were calculated (Appendix A). The Ka/Ks ratio is an important indicator reflecting the type and intensity of selection pressure during evolution, with Ka/Ks < 1 indicating negative selection pressure and Ka/Ks > 1 indicating positive selection pressure. Hence, the *HvSAD2*/*HvSAD13* gene pair (tandem duplication was displayed in a red box) with Ka/Ks = 2.55 underwent positive selection during evolution. The Ka/Ks of the *HvSAD4*/*HvSAD9* gene pair (segmental duplication) is 0.08, indicating that *HvSADs* possibly underwent purifying selection.

The syntenic relationships of *SADs* between barley and four other representative species (wheat, rice, maize, and Brachypodium distachyon) separately revealed 33, 6, 7, and 7 collinear gene pairs (Figure 7, Appendix A). The *SADs* have greater collinearity between barley and wheat and exhibit high homology with other monocotyledonous plants. Additionally, *HvSAD4* and *HvSAD9* exhibit multi-collinearity among species, indicating that they are more evolutionarily conserved.

### 2.7. Expression Profiles of HvSADs in Different Tissues

The expression data of 16 different growth stages and tissues of barley Morex were obtained from the BARLEX website, and clustered heatmaps of 13 *HvSADs* genes (The expression data of *HvSAD2* was absent on the website) were produced using TBtools v1.116 (accessed on 28 February 2023). Different colors and circle sizes indicated different expression levels (Figure 8). The expression patterns of genes are often related to their functions, and related analysis can provide clues for potential functional studies. The results indicated that *HvSADs* mainly exhibit three expression patterns in different tissues/organs. The first type was highly expressed during the reproductive growth stage, among the expression levels of *HvSAD4*, *HvSAD6*, *HvSAD7*, *HvSAD9*, and *HvSAD10* increased from the young developing inflorescences (5 mm) (INF1) to developing inflorescences (1–1.5 cm) (INF2) stage and further increased during the grain development stage. It is speculated that *HvSADs* may also be related to barley yield. The second type of *HvSADs* was mainly expressed during the nutrient growth stage, such as *HvSAD8* and *HvSAD11*, which were highly expressed at this stage but exhibited low expression at other tissue stages. Due to their obvious tissue specificity, these genes may be necessary for spike morphogenesis. The third type of *HvSADs* was almost not expressed in any tissue, including *HvSAD3*, *HvSAD5*, *HvSAD12*, and *HvSAD13*.

Rows represent *SAD* members, while columns show different developmental stages and tissues. The expression level of *SADs* [log10^(FPKM+1)^] is shown by the intensity of color, wherein blue blocks represent low expression and red blocks represent high expression. Different-sized circles describe the value of expression level, while blue circles indicate a value of zero. EMB, 4 day embryos; ETI, etiolated seedling, dark condition (10 DAP); LEA, shoots from seedlings (10 cm shoot stage); ROO1, roots from seedlings (10 cm shoot stage); ROO2: roots (28 DAP); INF1, young developing inflorescences (5 mm); INF2, developing inflorescences (1–1.5 cm); CAR5, CAR15: developing grain (5 DAP, 15 DAP); LEM, inflorescences, lemma (42 DAP); LOD, inflorescences, lodicule (42 DAP); PAL, dissected inflorescences, palea (42 DAP); EPI, epidermal strips (28 DAP); RAC, inflorescences, rachis (35 DAP); NOD, developing tillers, 3rd internode (42 DAP); SEN, senescing leaves (56 DAP).

### 2.8. Expression of 10 HvSADs under Abiotic Stress by qRT-PCR

To investigate the potential role of *HvSADs* in abiotic stress, barley seedlings at the third leaf stage were subjected separately to salt (200 mM), drought [20% (m/V) PEG6000], and cold (4 °C) stress treatments for 0, 4, and 12 h. Ten *HvSADs* with good primer specificity were selected for investigation, and qRT-PCR was used to detect the expression of genes in seedling leaves under three stress conditions. The results showed that most *HvSADs* exhibited significantly lower expression than the control for most of the time after drought, cold, and salt treatments, whereas the expression of most *HvSADs* was significantly upregulated under salt stress (Figure 9a–c). The selected *HvSADs* exhibited differential expression over time, but their response patterns varied under different stress conditions. Among them, *HvSAD4*, *HvSAD7*, and *HvSAD9* did not respond to cold stress but showed a significant upward and downward regulation trend under drought and salt treatments. Most notably, *HvSAD11* and *HvSAD12* showed highly significant upregulation under drought and salt stress treatments but a downregulation trend under cold stress. The above findings indicate that *HvSADs* respond to abiotic stresses and may participate in the response to environmental stress.

## 3. Discussion

Research on fatty acids has indicated that UFAs are essential and important nutrients for the human body. Additionally, UFAs are also an important component of plant cells, and so the study of their synthesis and functional mechanisms has important theoretical and practical significance. In plants, *SADs* have been cloned and characterized from *Arabidopsis* [28], peanut [29], cacao [8], cotton [30], and soybean [5], but there have been no reports on barley. Therefore, the identification and characterization of the *HvSADs* family at the genome-wide level will help elucidate their functional and evolutionary relationships in important crops. In this study, we comprehensively analyzed *HvSADs* and investigated their potential functions in development and abiotic stress responses.

In total, 14 *SADs* were identified in the barley genome and divided into four groups (Classes I–IV). Compared with cotton, tomato, and soybean, SADs from barley have a higher homology with wheat, rice, and maize, indicating that this gene family may potentially have more conservative evolution in monocotyledonous plants. The majority of *HvSADs* in the same group also showed relatively small changes in the exon–intron structure, motifs, and domain distribution, suggesting that *HvSADs* are more conserved within the species. The SADs in barley have relatively similar protein lengths and MWs among different groups, and it is similar in length and MW to SAD proteins in studied species such as *Arabidopsis* [28] and cocoa [8]. The FA_desaturase_2 domain of barley SADs has conserved histidine (EENRHG/DEKRHE) enriched regions [31], located in Motif2 and Motif1. Aspartic acid (D) and histidine (H) provide necessary binding sites and catalytic activity for Fe^2+^ HvSAD protein monomers [32]. The *HvSADs* in Group II have a higher proportion of introns compared to other members, and the retention of introns during transcription and translation may also increase compared to other members, thereby regulating gene expression and generating proteomic diversity, and even leading to phenotypic variation [33]. Interestingly, HvSADs (except HvSAD12) have chloroplast transit peptides, and some members also have mitochondrial transit peptides, which is consistent with the predicted localization results (Appendix A). This is different from cotton and oil tea, where HvSADs are not only localized in chloroplasts but also in mitochondria [30,34]. The above implies that HvSADs play a key role in plant photosynthesis and respiration. The existence of a link between plant photorespiration metabolism and fatty acid metabolism has been reported in soybean and *Arabidopsis* [35,36,37].

New genes mainly originate from gene duplication. Some duplicated genes undergo functionalization, which in plants encompasses varying degrees of functional differentiation and promotes the diversity of gene functions [38,39,40]. Analysis of the gene duplication events indicated that some *HvSADs* underwent tandem and segmental duplication processes in a similar way as the expansion of barley *GRAS* and *Arabidopsis* gene families [27,41]. Consequently, the expansion of the barley *SAD* family may also be the result of the combined effects of tandem and segmental duplication. Tandem duplication mainly occurs in the chromosomal recombination region, implying that recombination occurred in the regions where *HvSAD2* and *HvSAD13* are located. The Ka/Ks ratio of the segmentally duplicated *HvSAD* gene pairs was less than one. They underwent a purifying selection during the evolutionary process, avoiding detrimental mutations. This is crucial for the functional protection of the *HvSAD* family [42] (Appendix A). The collinearity analysis among different species showed that *HvSADs* have a high homology with Gramineae plants, especially wheat. *HvSAD4* and *HvSAD9* displayed multiple collinearities among species, providing potential clues for the origin and evolution of *HvSADs*. The conservation and functional diversity exhibited by the gene family during evolution enhances the adaptability of plants to various unfavorable survival environments, such as drought, disease, and temperature extremes, to ensure their growth and development [43,44].

The CREs themselves do not encode any proteins but typically bind to transcription factors to regulate the expression of adjacent genes. During plant growth, differentiation, and development, it is necessary to integrate signals from different tissues, development, and environments to regulate gene expression. Therefore, it is extremely important to focus on specific elements that regulate transcription initiation [45]. In the promoter regions, we identified a large number of CREs associated with hormone response, light response, and stress response, as well as a few growth and development elements. The number of light-responsive elements was the highest (182), followed by MeJA elements (CGTCAMotif/TGACG Motif, 84) and ABA reaction elements (ABRE, 69). It has been found that *Chlorella* significantly upregulated the transcription of *SAD* to induce the accumulation of total fatty acids, including oleic acid, under high light stress [46]. However, olives treated under dark conditions showed a significant 90-fold reduction in the level of *SAD1* and *SAD3* transcripts, but no noticeable difference in the total UFA content [47]. Hormones are also essential in regulating plant growth and development. MaABI5-like regulated ABA-induced cold tolerance by increasing the content of UFAs and flavonoids [48]. In *Arabidopsis*, the level of linolenic acid was increased through exogenous MeJA treatment to enhance drought and salt tolerance as well as cold tolerance [49]. Furthermore, MeJA participates in plant signal transduction, induces the synthesis of defensive compounds, and plays a great part in plant resistance to stresses and diseases [44,50]. The above results emphasize that utilizing CREs that potentially affect plant development and environmental response can be used to promote crop improvement.

In some plants, *SADs* have been found to have tissue and stage-specific expression. *AhSAD3* in peanuts was mainly expressed in developing seeds in accordance with the oil accumulation stage, while *AhSAD1/2* was expressed in developing seeds, leaves, stems, roots, and flowers, and the expression level in the flowers was even higher than that in the seeds [29]. In *Arabidopsis*, the double mutations in both *FAB2* and *AAD5*, two SAD-encoding genes, caused embryo development to stop before the globular stage, and these two genes along with *AAD1*, another *SAD*-encoding gene, participated in the formation of embryonic keratin in the later stage of seed growth [7]. *OeSAD1* was expressed most highly in olive leaves and least in the flowers, while *OeSAD2* was mainly expressed in the fruit peel and was lowest in the leaves, exhibiting a significant inducing effect in the fruit [51]. This study showed that *HvSADs* are not only expressed in a specific tissue type. For example, *HvSAD4* and *HvSAD9* were expressed at higher levels in each tissue compared to other members, suggesting that they have a regulatory effect on multiple development stages. However, *HvSADs* also exhibited high expression levels in specific tissue types, such as *HvSAD10* being most expressed in CAR15, which is consistent with the research results in peanuts.

Existing studies have shown that SADs are associated with plant abiotic stress resistance, particularly with a crucial impact on cold stress resistance [10,47,52,53,54,55]. In tea trees and flax, the expression of the *SADs* was strongly induced by cold and drought stresses [52,55]. It is noteworthy that in addition to most ARE and GC motifs, the *HvSADs* also had CCAAT-box (MYBHv1 binding site), MBS (MYB binding site involved in drought-inducibility), LTR (low-temperature responsiveness), TC-rich repeats (defense and stress responsiveness), and MBSI (MYB binding site involved in the regulation of flavonoid biosynthesis genes) elements, which offer the possibility of the *HvSADs* to respond promptly to drought, cold, and salt stresses. [56,57] In the study, *HvSADs* were also found to respond to cold, drought, and salt stresses, but each member responded to different stresses in different ways. Low-temperature stress often leads to the fixation of membrane proteins, and the content of UFAs in plants increases to a certain extent in order to maintain the fluidity of the cell membrane [47]. *HvSAD8* was significantly upregulated by cold stress, with the highest expression level in the aboveground parts of the seedlings, implying that it is important for the protection of barley seedlings against severe cold. The expression levels of *HvSAD5* and *HvSAD10* were significantly downregulated and then upregulated under cold stress, indicating a rapid response to stress and controlled through negative regulation. In addition to positively responding to cold stress, studies have also found that increasing the content of UFAs in plant membrane lipids can greatly enhance the tolerance of the photosynthetic machinery to salt stress. In *Arabidopsis*, salt stress significantly reduced the expression of *AtACP5*, and knocking out the gene increased the sensitivity of plants to salt stress. The overexpression of *AtACP5* led to changes in FA composition, thereby improving salt tolerance [58,59]. Similar to other abiotic stresses, plants resist drought stress by regulating the content of UFAs [60,61]. *HvSAD11* and *HvSAD12* strongly responded to drought and salt stress and were significantly upregulated after 4 h and 12 h, exhibiting greater upregulation under these stresses compared to cold stress. This may be related to the fact that, compared to other cereal crops, barley itself has strong salt, drought, and cold tolerance. The *SAD* family in plants influences lipid synthesis, seed development, and stress resistance, which has positive implications for genetic improvement and breeding in plants [53,62,63].

## 4. Materials and Methods

### 4.1. Identification of SADs in Barley

To find candidate *SADs* in the barley genome, the hidden Markov model profile of the PF03405 (FA_desaturase_2 domain) file was downloaded from the Pfam database (http://pfam-legacy.xfam.org/) (accessed on 24 August 2022) [64,65], and the protein sequences with this domain were screened using TBtools v1.098669 (accessed on 24 August 2022) (E-value ≤ 10^−5^) [66]. Meanwhile, for reliable results, the identified sequences of seven SADs of *Arabidopsis* were also used as reference sequences, and a BLASTP search was performed with the barley genome in the Ensembl Plants database (http://plants.ensembl.org/index.html) (accessed on 24 August 2022) as well as in the BARLEX database (http://barlex.barleysequence.org) (accessed on 24 August 2022), with an E-value ≤ 10^−5^ and identity ≥ 50% set as the filtering conditions to obtain the potential *HvSAD* members [28,67,68]. Using NCBI (https://www.ncbi.nlm.nih.gov) (accessed on 25 August 2022) and SMART (http://smart.embl-heidelberg.de/) (accessed on 25 August 2022) online software, domain validation was performed on the screened barley SAD protein sequences [69]. Next, the incomplete reading frame and short and redundant sequences were removed manually, after which the *SAD* gene and protein sequences in barley were finally obtained.

### 4.2. Physicochemical Properties Analysis

The composition and physical and chemical characteristics of the HvSADs were analyzed with ExPASy (https://web.expasy.org/) (accessed on 1 September 2022), and the subcellular localizations of the HvSADs were assessed using WoLF PSORT (https://wolfpsort.hgc.jp/) (accessed on 1 September 2022) [70,71]. Signal and chloroplast transit peptides were predicted using Ipsort (https://ipsort.hgc.jp/) (accessed on 1 September 2022) [72]. Detailed information on all HvSADs is provided in Table 1.

### 4.3. Phylogenetic Analysis

A phylogenetic tree of SADs among eight plant species was constructed using the neighbor-joining (NJ) method in MEGA7.0 software. The bootstrap value was set to 1000, and Evolview (http://www.evolgenius.info/evolview/#/) (accessed on 7 March 2023) was used to display the evolutionary tree [73,74]. Additionally, to study the characteristic structural domains of barley SAD members, multiple sequence comparisons of HvSAD protein sequences were performed using DNAMAN version: 9.0.1.116 (accessed on 7 March 2023).

### 4.4. Analysis of Conserved Motif and Gene Structure

The exon-intron structures of *HvSADs* were examined using TBtools and the GFF3 data (Generic Feature Format version 3 Data) from Ensembl Plants. The conserved motifs of HvSADs performed a thorough investigation with MEME (http://meme-suite.org/) (accessed on 12 February 2023), with a maximum of 10 motifs and motif width between 6 and 50 amino acid residues. Finally, the full graphics of the conversed motif and gene structure were visualized by the BioSequence Structure Drawers function of TBtools.

### 4.5. Cis-Regulatory Elements Prediction in the Promoter Regions of HvSADs

For each *HvSAD*, a 2000 bp sequence upstream of the initiation codon (ATG) was extracted using TBtools v1.116 (accessed on 12 February 2023) and submitted to the PlantCARE website (http://bioinformatics.psb.ugent.be/webtools/plantcare/html/) (accessed on 12 February 2023) to obtain information on CREs [75]. The obtained prediction information was categorized and summarized. Eventually, the CREs were displayed using TBtools and Excel 2016 software (Microsoft Corp., Redmond, WA, USA).

### 4.6. Chromosome Distribution, Gene Duplication, and Selective Pressure Analysis

Genomic data of barley, wheat (*Triticum aestivum*), rice (*Oryza sativa*), maize (*Zea mays*), and *Brachypodium distachyon* were downloaded from the Ensembl Plants database. The TBtools v1.116 (accessed on 9 February 2023) was used to analyze and visualize the chromosomal distribution, gene duplication events within the barley genome, as well as the gene level collinearity among plant genomes. The Simple Ka/Ks Calculator (NG) tool in TBtools v1.116 (accessed on 9 August 2023) was used to calculate the ratio of non-synonymous to synonymous substitution (Ka/Ks) for duplicated gene pairs.

### 4.7. Expression Analysis of HvSADs Members

The expression levels of *HvSADs* in different tissues and development stages were downloaded from the transcriptome data of RNA sequence from the BARLEX database. The gene expression values are represented by fragments per kilobase of exon per million fragments mapped (FPKM). A heatmap of the expression pattern was analyzed by TBtools to analyze the expression of each *SAD* in different tissues of barley.

### 4.8. Plant Material, Stress Treatment, RNA Extraction, and qRT-PCR Analysis

After disinfecting the barley Morex seeds, they were germinated under dark conditions at 24 °C. Seedlings exhibiting strong and consistent growth were selected for hydroponic cultivation and were cultivated with 1/2 Hoagland’s nutrient solution in an artificial illumination incubator under a day/night temperature of 24 °C/22 °C with 14 h of 100% of full light and 10 h of darkness. The nutrient solution was changed every three days. At the third-leaf stage, the seedlings were treated under low-temperature stress (4 °C), salt stress (200 mM NaCl), and simulated drought stress [20% polyethylene glycol (PEG)6000] separately. The control group was cultivated with a nutrient solution only. The leaves were, respectively, sampled at 0 h, 4 h, and 12 h and stored at −80 °C after rapidly freezing with liquid nitrogen.

Total RNA was extracted using a Plant RNA Extraction Kit (Promega, Beijing, China) and then reversely transcribed to cDNA by a Reverse Transcription Kit (YEASEN, Shanghai, China), all according to the reagent instructions. The quantitative real-time PCR (qRT-PCR) amplification was performed using Hieff^®^ qPCR SYBR Green Master Mix (YEASEN, Shanghai, China) in a CFX96 Touch^TM^ Real-Time PCR Detection System (Bio-Rad, Hercules, CA, USA). Primers were designed according to the CDS sequences of *HvSADs* using Primer Premier 5 software (Premier Biosoft Interpairs, Palo Alto, CA, USA) with the following settings: PCR product size for 100–300 bp, primer length for 22 ± 3 bp. Their specificity was then confirmed by using the Ensembl Plants database for BLASTN search in the barley genome. *HvActin (HORVU. MoreX.r3.1HG0003140)* was used as the internal reference gene. The sequence information of all primers is listed in Appendix A. Three biological replicates and three technical replicates were tested for each sample, and the relative expression levels were calculated using the 2^−ΔΔCt^ method [76]. Gene expressions for selected *HvSADs* were analyzed using GraphPad Prism version 9.5.0 for Windows (GraphPad Software, San Diego, California USA, www.graphpad.com).

## 5. Conclusions

This study identified 14 *HvSADs* genes from the barley genome. Based on a phylogenetic and gene structure analysis, the HvSADs were classified as Class I to Class IV. Members of the same group exhibited similar structural characteristics. Gene replication analysis indicated that tandem duplication, fragment duplication, and purification selection contributed to the expansion and evolution of the *HvSAD* family. In addition, a commonality analysis between barley, rice, wheat, maize, and *Brachypodium distachyon* showed varying degrees of correlation. The *HvSADs* were found to be influenced by various factors, including hormones, light, and stress. In addition, the *HvSADs* were highly expressed during young ear and seed development and may be involved in the regulation of reproductive growth. Several *HvSADs* exhibited different expression patterns under abiotic stress. Overall, the genome-wide identification and molecular characterization of *HvSADs* provide new insights into their evolutionary history and functional roles, contributing to a comprehensive analysis of the regulatory mechanisms and genetic improvement of plant oil biosynthesis and metabolism.

## Figures and Tables

**Figure 1 ijms-25-00113-f001:**
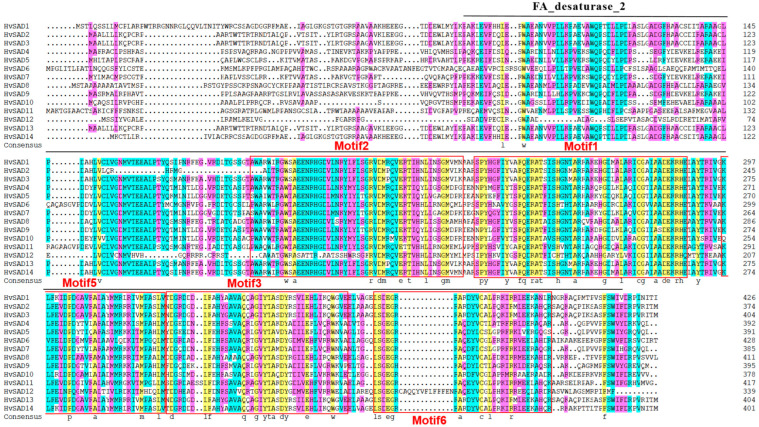
Multiple sequence alignment of HvSADs. Different colors represent different proportions of homology. Yellow represents = 100%, blue represents ≥ 75%, and purple represents ≥ 50%; the black solid line represents the FA_desaturase_2 domain. The range shown in the red box is the common motifs by HvSADs.

**Figure 2 ijms-25-00113-f002:**
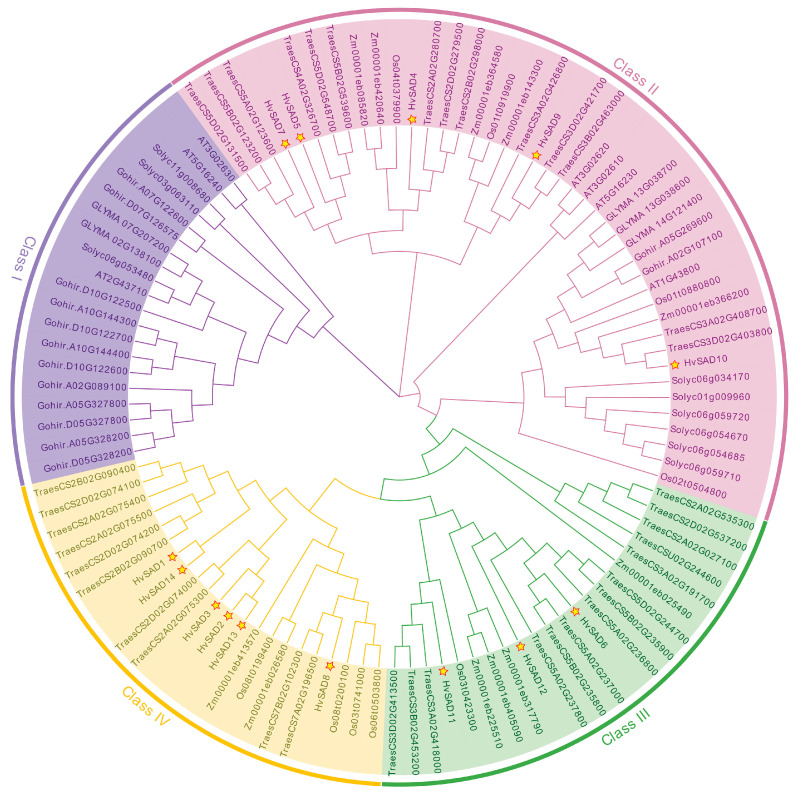
Phylogenetic analysis of HvSADs and their homologs from various organisms. AT represents *A. thaliana*, Zm represents *Z. mays*, Os represents *O. sativa* L, Traes represents *T. aestivum*, GLYMA represents *G. max*, Gohir represents *G. hirsutum*, Soly represents *S. lycopersicum*, and Hv represents *H. vulgare*. The phylogenetic tree was generated using the NJ method by MEGA7; bootstrap values = 1000. SADs are classified into four groups (Class I–IV) and are distinguished by different colors. HvSADs are marked with stars.

**Figure 3 ijms-25-00113-f003:**
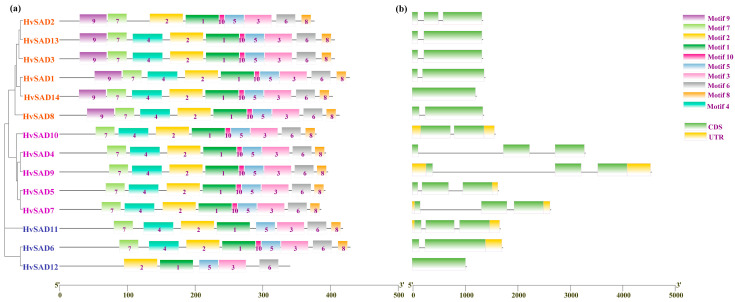
Conserved motifs and gene structures. (**a**) Motif patterns of HvSADs. Boxes with different colors represent various conserved motifs. (**b**) Exon–intron architectures of *HvSADs*. The introns, exons, and untranslated regions are represented by black lines, green boxes, and yellow boxes, respectively.

**Figure 4 ijms-25-00113-f004:**
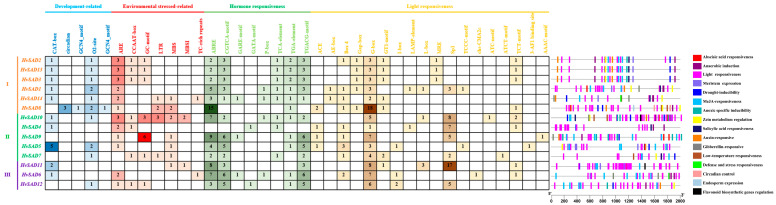
CREs distribution in the promoter regions of *HvSADs*. The CREs in the promoter of each *HvSAD* were classified based on the putative functions, including hormone-responsive CREs, light-responsive CREs, environmental stress-responsive CREs, and growth and development-responsive CREs. The positional distribution of various CREs on promoters is shown as rectangles with different colors. The number in each box represents the quantity of each CRE for the corresponding gene.

**Figure 5 ijms-25-00113-f005:**
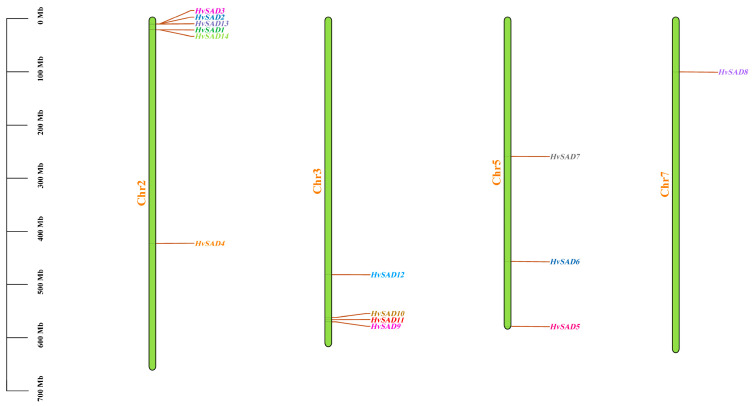
Chromosome distribution of *HvSADs*. Chromosome numbers are shown at the left of each chromosome, and the name of each *HvSAD* is labeled on the right side of each chromosome. The scale bar on the left side indicates the chromosome lengths (Mb).

**Figure 6 ijms-25-00113-f006:**
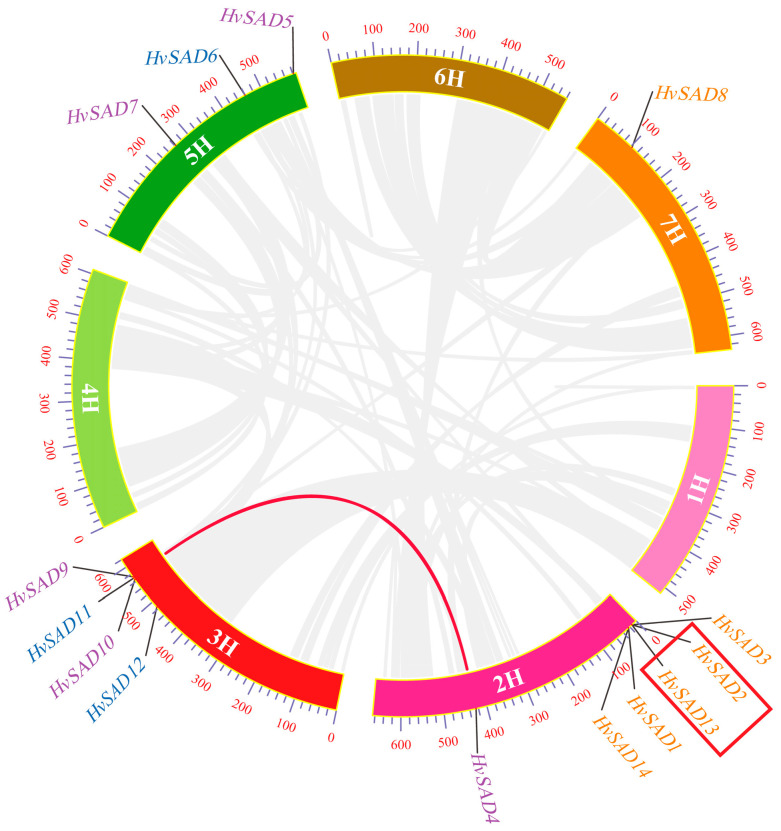
Gene duplications of *HvSADs*. Segmentally duplicated *HvSAD* gene pairs are linked by the red lines between chromosomes. Segmentally duplicated gene pairs within the rice genome are linked by the gray lines. The tandem duplication gene pairs are displayed in a red box. The chromosome numbers are shown at the center of each chromosome. The scale bar marked on the chromosome indicates chromosome lengths (Mb).

**Figure 7 ijms-25-00113-f007:**
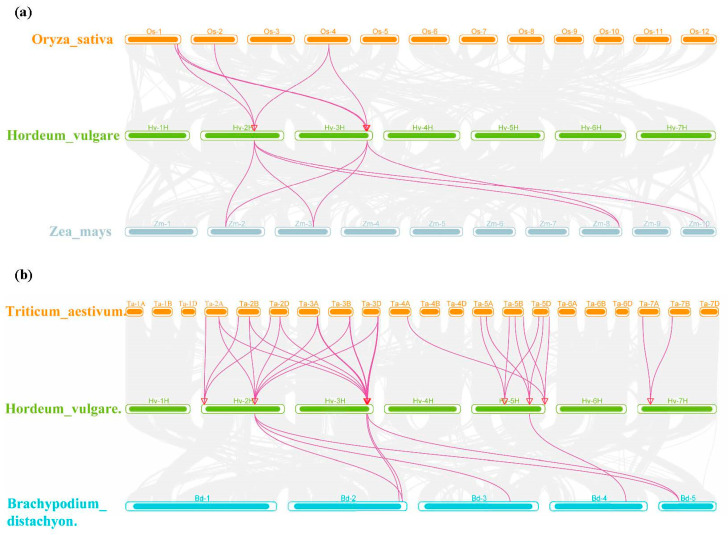
Collinearity relationship of the *SADs* in barley and four other plant species, including rice and maize (**a**), wheat, and Brachypodium distachyon (**b**). Gray lines in the background indicate the syntenic blocks between barley and other plant genomes, while the purple lines highlight the syntenic *HvSAD* gene pairs and red triangles show gene locations.

**Figure 8 ijms-25-00113-f008:**
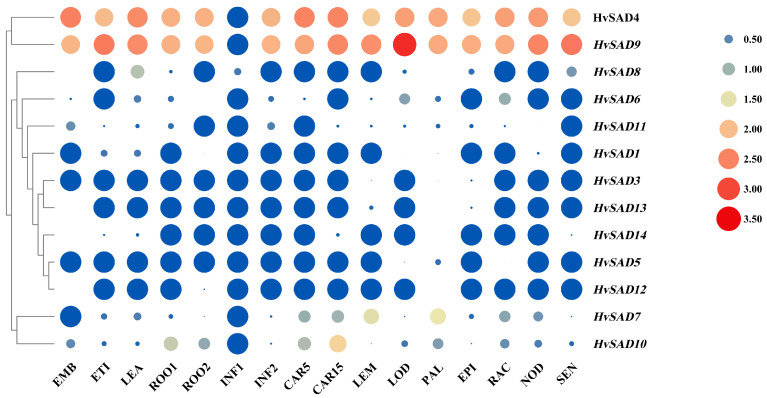
Expression profiles of *HvSADs* in different tissues and development stages.

**Figure 9 ijms-25-00113-f009:**
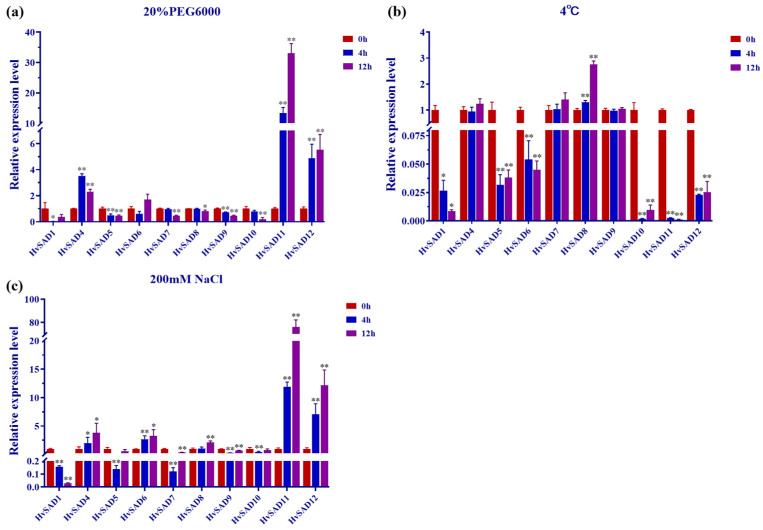
Expression analysis of 10 selected *HvSADs* under abiotic stresses using qRT-PCR, *viz*. drought (**a**), cold (**b**), salt (**c**) stresses. The *y*-axis represents the relative expression level of the *HvSAD* gene in plant leaves based on qRT-PCR analysis. The *x*-axis represents the analyzed *HvSADs*, with the red column representing a treatment time of 0 h, the blue column representing 4 h, and the purple column representing a treatment time of 12 h. A *t*-test was used to analyze significant differences in data, “*” represents *p* < 0.05, and “**” represents *p* < 0.01. Error bars represent standard deviations of the means from three technical replicates.

**Table 1 ijms-25-00113-t001:** Detailed information of the identified barley SADs.

Gene Name	Gene ID	Protein					Subcellular Localization	Domain	Chromosome	Genomic Location
		Size (aa)	MW (Da)	pI	Aliphatic Index	GRAVY				
*HvSAD1*	HORVU.MOREX.r3.2HG0106740	427	47,347	8.47	85.55	−0.267	Chloroplast	FA_desaturase_2	Chr2	23,954,883−23,956,257
*HvSAD2*	HORVU.MOREX.r3.2HG0101470	375	41,677.74	8.68	87.33	−0.257	Mitochondrial	FA_desaturase_2	Chr2	13,191,866−13,193,197
*HvSAD3*	HORVU.MOREX.r3.2HG0101440	405	44,883.21	7.19	84.96	−0.241	Mitochondrial	FA_desaturase_2	Chr2	13,111,139−13,112,470
*HvSAD4*	HORVU.MOREX.r3.2HG0161410	392	44,559.57	6.04	73.44	−0.490	Chloroplast	FA_desaturase_2	Chr2	425,614,894−425,618,180
*HvSAD5*	HORVU.MOREX.r3.5HG0535350	391	44,667.93	6.05	80.36	−0.405	Chloroplast	FA_desaturase_2	Chr5	581,725,723−581,727,357
*HvSAD6*	HORVU.MOREX.r3.5HG0486420	428	47,617.36	7.59	73.93	−0.306	Chloroplast	FA_desaturase_2	Chr5	459,736,400−459,738,111
*HvSAD7*	HORVU.MOREX.r3.5HG0457600	385	43,660.82	5.97	74.75	−0.381	Chloroplast	FA_desaturase_2	Chr5	262,136,160−262,138,788
*HvSAD8*	HORVU.MOREX.r3.7HG0668240	412	45,390.46	6.87	72.91	−0.344	Chloroplast	FA_desaturase_2	Chr7	103,262,598−103,263,947
*HvSAD9*	HORVU.MOREX.r3.3HG0310210	395	44,530.92	6.65	78.61	−0.409	Chloroplast	FA_desaturase_2	Chr3	573,217,384−573,221,932
*HvSAD10*	HORVU.MOREX.r3.3HG0307490	378	42,389.33	6.24	79.29	−0.324	Chloroplast	FA_desaturase_2	Chr3	564,864,759−564,866,328
*HvSAD11*	HORVU.MOREX.r3.3HG0309010	417	45,352.59	6.87	78.01	−0.177	Chloroplast	FA_desaturase_2	Chr3	569,309,951−569,311,619
*HvSAD12*	HORVU.MOREX.r3.3HG0288610	339	38,752.42	9.03	76.31	−0.297	Cytoplasmic	FA_desaturase_2	Chr3	484,477,447−484,478,466
*HvSAD13*	HORVU.MOREX.r3.2HG0101500	405	44,883.21	7.19	84.96	−0.241	Mitochondrial	FA_desaturase_2	Chr2	13,265,580−13,266,911
*HvSAD14*	HORVU.MOREX.r3.2HG0106840	402	44,563	8.4	85.55	−0.236	Chloroplast	FA_desaturase_2	Chr2	24,197,160−24,198,368

## Data Availability

Data is contained within the article and Appendix A.

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
