# Peer review of "Genome-Wide Identification and Expression Analysis of the Stearoyl-Acyl Carrier Protein Δ9 Desaturase Gene Family under Abiotic Stress in Barley"

_ijms, 2023, doi:10.3390/ijms25010113_

Round 1

Reviewer 1 Report

Comments and Suggestions for Authors

I am providing feedback for the manuscript "Genome-wide identification and expression analysis of the SAD gene family under abiotic stress in barley". The authors studied the SAD gene family in barley based on seven homologous starting SAD genes from Arabidopsis. I have provided my suggestions and comments in the pdf file to improve the manuscript's visibility and interest in the scientific community. Please address the comments and resubmit the manuscript.

Comments on the Quality of English Language

It is important to review the manuscript for any grammar and word choice errors. e.g. line 150

Author Response

Thanks!  Please find the attached file.

Reviewer 2 Report

Comments and Suggestions for Authors

Dear authors,

your manuscript presents a bioinformatics study combined with an analysis of the expression of desaturase genes in barley under different types of stress.

Understanding the importance of studying this crop and the role of desaturases in plant adaptation to stressors, I highly appreciate your research.

Still, I have some minor comments. Would it be nice to indicate under what lighting conditions did the barley plants grow in the hydroponic system? Was the medium aerated to prevent root hypoxia? How often was Hoagland's solution renewed when growing plants?

I would like a more in-depth discussion of the relationship between the structure of the regulatory regions of deseturase genes and specific or nonspecific responses to various stresses.

I am confident that it will not be difficult for you to make the necessary changes.

Author Response

Thanks!
